# Sex Differences in Glutathione Peroxidase Activity and Central Obesity in Patients with Type 2 Diabetes at High Risk of Cardio-Renal Disease

**DOI:** 10.3390/antiox8120629

**Published:** 2019-12-07

**Authors:** Mia Steyn, Karima Zitouni, Frank J Kelly, Paul Cook, Kenneth A Earle

**Affiliations:** 1Thomas Addison Unit: St George’s University Hospitals NHS Foundation Trust, London SW17 0QT, UK; mia.steyn@gstt.nhs.uk; 2Institute of Infection and Immunity, St George’s University of London, London SW17 0QT, UK; kzitouni@sgul.ac.uk; 3NIHR Health Protection Research Unit in Health Impact of Environmental Hazards, King’s College London, London SE1 9NH, UK; frank.kelly@kcl.ac.uk; 4Chemical Pathology and Metabolic Medicine, University Hospital Southampton NHS Foundation Trust, Southampton SO16 6YD, UK; paul.cook@uhs.nhs.uk; 5Institute of Medical and Biomedical Education, St George’s University of London, London SW17 0QT, UK

**Keywords:** Type 2 diabetes, oxidative stress, antioxidants, glutathione peroxidase, sex differences, cardiovascular risk

## Abstract

Women with type 2 diabetes (T2DM) have an increased susceptibility of developing cardio-renal disease compared to men, the reasons and the mechanisms of this vulnerability are unclear. Since oxidative stress plays a key role in the development of cardio-renal disease, we investigated the relationship between sex, plasma antioxidants status (glutathione peroxidase (GPx-3 activity), vitamin E and selenium), and adiposity in patients with T2DM at high risk of cardio-renal disease. Women compared to men had higher GPx-3 activity (*p* = 0.02), bio-impedance (*p* ≤ 0.0001), and an increase in waist circumference in relation to recommended cut off-points (*p* = 0.0001). Waist circumference and BMI were negatively correlated with GPx-3 activity (*p* ≤ 0.05 and *p* ≤ 0.01, respectively) and selenium concentration (*p* ≤ 0.01 and *p* ≤ 0.02, respectively). In multiple regression analysis, waist circumference and sex were independent predictors of GPx-3 activity (*p* ≤ 0.05 and *p* ≤ 0.05, respectively). The data suggest that increased central fat deposits are associated with reduced plasma antioxidants which could contribute to the future risk of cardio-renal disease. The increased GPx-3 activity in women could represent a preserved response to the disproportionate increase in visceral fat. Future studies should be aimed at evaluating if the modulation of GPx-3 activity reduces cardio-renal risk in men and women with T2DM.

## 1. Introduction

Worldwide, the incidence of cardiovascular disease (CVD) is the main cause of premature death in patients with type 2 diabetes, which is more common in women than their male counterparts [1,2]. Excess body weight has attracted much attention as one of the contributing risk factors to CVD, especially with the continuous global increase, and recent epidemiological data suggest that the accumulation of central fat deposits in women contributes to this excess risk of CVD [3,4]. Clinical studies have shown strong correlations between markers of oxidative stress and obesity [5], with central fat deposits acting as a source of pro-inflammatory cytokines and reactive oxygen species (ROS) [6,7].

A strong experimental evidence base shows that the dysregulated production of ROS is central to the pathogenesis of diabetes and its vascular complications [8,9]. Reactive oxygen species have an impact on structural vascular remodeling by causing a loss of elasticity in the arterial wall [10], and impairing endothelial function by reducing the bioavailability of nitric oxide [11,12]. Some authors have reported that a greater dependency on the availability of nitric oxide for vascular responsiveness occurs in women than men with diabetes who are at risk of renal disease [13]. Moreover, evidence is accumulating that endogenous pathways involving antioxidants play a key role in limiting the effect of ROS on the development of CVD and renal disease in diabetes [9,14,15]. 

The antioxidant defense system is composed of different antioxidant molecules that work at different levels and milieus to neutralize free radicals. The first line of antioxidant defense against oxidative stress that removes ROS are the three antioxidant enzymes, superoxide dismutase (SOD), glutathione peroxidase (GPx), and catalase. SOD catalyzes the dismutation of the superoxide anion (O^2^_*_) into hydrogen peroxide (H_2_O_2_) intracellularly and extracellularly. Catalase and GPx both remove hydrogen peroxide, while GPx can also scavenge lipid hydroperoxides to alcohols [16]. A meta-analyses of observational cohort and case-control studies reported an inverse relationship between coronary heart disease with circulating levels of superoxide dismutase, glutathione peroxidase, and catalase activities [17].

In this research paper, we study cardio-renal disease vulnerability, hence, the focus is on the antioxidant enzyme GPx and, specifically, one of the seven isoforms, which is plasma GPx (GPx-3), because it is mainly derived from the renal tubular epithelium [18], although it is expressed in other organs, such as the lung, liver, heart, eyes, and adipose tissue [19,20,21]. Glutathione peroxidase anti-oxidative activity in the plasma is mostly owed to the GPx-3, which is a selenoprotein enzyme containing selenocysteine at its active site. GPx-3 activity is reduced in association with increased oxidative stress in various degrees of renal failure [22]. Crawford et al. (2011) reported a direct relationship between GPx-3 activity and kidney function deterioration over one year in patients with chronic kidney disease (CKD) [23]. Interestingly, the progression of CKD is associated with a two- to four-fold increased risk of dying prematurely from CVD, which is high in female diabetic patients [1]. Serum hydroperoxides levels as an index of oxidative stress were observed to be different between male and female participants [24]. Therefore, we seek to investigate whether GPx-3 activity and its co-factor selenium and vitamin E status are differentially affected by sex and how it relates to measures of obesity in patients with type 2 diabetes who are free from, but at high risk of, developing cardiovascular and renal complications.

## 2. Materials and Methods

We studied a cohort of 170 adult men and women with type 2 diabetes mellitus who were recruited as part of the PREVENT trial from general practices in South West London, UK (protocol previously published, Trial Registration ISRCTN 97358113 [25]. Written informed consent was obtained from all participants prior to inclusion in the study and ethical permission was granted by the National Health Service Ethics Research Committee (05/Q0803/57). Type 2 diabetes mellitus was diagnosed according to World Health Organization (WHO) criteria. Eligible patients also had hypertension (3 consecutive sitting systolic blood pressure readings >140 mmHg and/or diastolic >90 mmHg without treatment, or were receiving treatment for hypertension) and early chronic kidney disease (CKD), defined as eGFR <90 and >45 mL/min/1.73 m^2^ and/or a urinary albumin:creatinine ratio >3 mg/mmol. Patients were excluded if they had any of the following: history of cardiovascular disease (defined as a clinical record of ischemic heart disease, cerebrovascular disease or peripheral vascular disease); history of malignancy or other life-threatening illness, current pregnancy, systolic blood pressure >200 mmHg or diastolic blood pressure >160 mmHg, hemoglobin A1c (HbA1c) > 10% (86 mmol/mol) or nephrotic range proteinuria (total protein excretion rate >3 g/day or an albumin: creatinine ratio >300 mg/mmol). Family history, along with personal medical history, smoking status, and treatment histories were recorded on electronic proformas. 

### 2.1. Biochemical Assessments

Venous blood was sampled after a 12-h overnight fast at entry into the study. Plasma creatinine was measured using an isotope dilution mass spectrometry reference measurement procedure and was used to estimate renal function using the Chronic Kidney Disease Epidemiology Collaboration (CKD-EPI) equations [26]. Three early morning urine samples were collected to measure the urinary albumin:creatinine ratio. The total triglycerides and total- and high-density lipoprotein (HDL)-cholesterols were estimated using an enzymatic assay (Roche system 702 on Cobas 8000/702; Roche Diagnostics, Mannheim, Germany).

### 2.2. Antioxidant Defense Assessment

Plasma glutathione peroxidase activity was measured using a coupled assay system based on the oxidation of reduced glutathione (GSH) leading to the generation of NADP+ which was monitored at 340 nm [27].

Vitamin E as plasma α -tocopherol was analyzed by a high-performance liquid chromatography system with ultraviolet detection using tocopherol acetate as the internal standard. α-Tocopherol concentration in the samples was calculated by relating their peak areas to that of the internal standard. The plasma contents of absolute and lipid standardized vitamin E was expressed relative to cholesterol [27].

Plasma selenium (Se) was analyzed at Southampton University Hospital Trace Element Laboratory by inductively coupled plasma mass spectrometry NexION 300D (Perkin-Elmer, Beaconsfield, UK). ^78^Se was measured using ammonia (0.5 mL/min) as the dynamic reaction cell gas to remove argon based isobaric interferences [28]. Samples were run against a Sigma Serum calibration curve, which was spiked with different concentrations of Se standard reference solution 1000 ppm (Fisher Chemical, Loughborough, UK). Samples were diluted with a diluent containing 0.5% butan-1-ol for calibration, test, and quality control, which increases the ionization of Se and the sensitivity of the signal [29]. Rhodium was used as the internal standard. Internal quality control material was used throughout the assay, observed values were 0.60 ± 0.04 µmol/L (target 0.55 µmol/L), 1.16 ± 0.10 µmol/L (target 1.15 µmol/L) and 2.08 ± 0.11 µmol/L (target 2.10 µmol/L).

### 2.3. Body Adiposity Measurements

Anthropometric measures included height in meters, weight in kilograms, and waist circumference in centimeters. Body mass index (BMI) was calculated as weight in kilograms divided by height in square meters. Systolic blood pressure (SBP) and diastolic blood pressure (DBP) were measured in the sitting position. The total body fat percentage was assessed as bio-impedance. Briefly, patients were asked to stand without shoes or outer garments on their feet on a bio-impedance foot-to-foot analyzer (Tanita BF-350, Tanita Corporation, Tokyo, Japan).

### 2.4. Statistical Analysis

Data were analyzed using SPSS (IBM Corp. Released 2013. IBM SPSS Statistics for Windows, Version 22.0. Armonk, NY: IBM Corp.) Sex-specific descriptive analyses of our study cohort’s characteristics and cardiovascular risk factors was performed. Continuous variables were expressed as mean ± standard deviation (SD) and categorical variables were expressed as proportions. Normal distribution was assessed using visual inspection and Shapiro-Wilk tests of normality, with significance of >0.05 accepted as indicating normal distribution. Data with skewed distribution were expressed as median (inter-quartile range) and were log-transformed before statistical analyses. Where log transformation was not possible, data were analyzed using non-parametric tests. Statistical significance was defined as *p* ≤ 0.05. Sex differences were assessed using the two-sample t-test. Bivariate relationships between clinical and biochemical variables was analyzed using Pearson and Spearman’s Rho correlation analysis, followed by regression analysis using generalized linear models with GPx-3 activity as the dependent variable. 

## 3. Results

The data from 170 patients, whose clinical characteristics are shown in Table 1, were analyzed. There were no significant differences between the male and female patients in chronological age, duration of diabetes, smoking status, blood pressure, and renal function. Women had significantly higher total-, HDL-, and LDL-cholesterols than men. The proportion of men prescribed HMGCoA reductase inhibitors and antihypertensive agents was higher (79% vs 68% and 79% vs 70%, respectively) compared with the women, but this difference did not reach statistical significance. There were no differences in the use of oral hypoglycemic agents and/or insulin in the treatment of diabetes or the quality of diabetes control according to the glycated hemoglobin level. 

Body mass index and waist circumference were similar in both sexes. The relative increase in waist circumference, when compared to the recommended cut off-points of the International Diabetes Federation [30] and bio-impedance assessment of total body fat was significantly greater in women (Table 1).

Plasma levels of vitamin E, selenium, and GPx-3 activity were assessed as markers of antioxidant defense. GPx-3 activity was significantly higher in women in comparison to men (Table 2 and Figure 1). There was a strong correlation between GPx-3 activity and selenium concentration (*R* = 0.305, *p* ≤ 0.001).

In the whole group, waist circumference and BMI both significantly negatively correlated with GPx-3 activity (*r* = −0.168; *p* ≤ 0.05 and *r* = −0.182; *p* ≤ 0.01, respectively) and selenium concentration (*r* = −0.212; *p* ≤ 0.01 and *r* = −0.184; *p* ≤ 0.02, respectively) but not with vitamin E. Bio-impedance had no relationship with either GPx-3 activity, vitamin E, or selenium levels (*r* = 0.01, *p* = 0.09; *r* = −0.04, *p* = 0.6; *r* = −0.08, *p* = 0.3, respectively).

In male patients, GPX-3 activity correlated negatively with BMI (*r* = −0.245, *p* ≤ 0.05), while there was no relationship between GPX-3 activity and BMI in the female cohort.

Linear regression models were used to assess the impact of sex on the above associations. In model A, GPx-3 activity was used as the dependent variable, waist circumference and sex were entered as the independent variables, these were strong predictors of GPx-3 activity (β = −1.48; *p* ≤ 0.05 and β = 41.0; *p* ≤ 0.05, respectively). In Model B, waist circumference was replaced by bio-impedance in model A, sex remained a predictor of GPx-3 activity (β = 58.01, *p* = 0.01) (Table 3). 

We run similar regression models to model A and B using vitamin E as the dependent variable, body adiposity measures and sex as independent variables, the analyses did not show any interactions.

## 4. Discussion

Obesity is a major and a modifiable determining factor of cardiovascular disease that has recently been recognized as a potential promoter of renal disease in women. In this study, we found that female patients with type 2 diabetes at high risk of progressive cardio-renal disease have a relatively greater increase in central obesity and a surprisingly elevated activity of GPx-3 compared with men. While increased waist circumference was associated with lower GPx-3 activity and its co-factor selenium in the whole population. No associations were found between total body fat and vitamin E.

The NHF Risk Factor Prevalence Study of 4487 women with no history of heart disease, diabetes, or stroke unveiled that central obesity markers are far more superior than BMI in predicting women who are at risk of CVD [4]. This is in line with our study findings in which BMI did not show any sex difference, however, there was a striking difference in central obesity as defined by the ratio of waist circumference to recommended cut-off points of the International Diabetes Federation and bio-impedance.

Experimental studies have shown that obesity related oxidative stress may activate select signaling pathways in the liver, leading to lipogenesis and steatosis, and thereby exacerbates insulin resistance disease progression and contributes to cardiovascular complications [31]. Adipocytes produce a wide variety of metabolically active substances and dysregulation of these cause oxidative stress, which contribute to obesity-related complications [5]. Whilst fat accumulates in the body, the production of reactive oxygen species including superoxide anion increases [5]. Superoxide anion is dismutated by superoxide dismutase and as a result, hydrogen peroxide is produced, the latter, is converted to water by the anti-oxidative activity of glutathione peroxidase. Lower SOD and GPx activities have been reported in obese and overweight women and men with metabolic syndrome without diabetes compared to normal weight controls [32,33]. Furthermore, others have reported lower levels in patients with diabetes compared to non-diabetic control subjects, the magnitude of which was greater in the presence of obesity [34].

On the one hand, from the above-mentioned studies [5,32,33,34], it is apparent that there is an increased oxidative stress in the presence of obesity as a result of diminished antioxidant enzyme activities. On the other hand, there is a paradox with the current study that women have increased GPx-3 activity even though they have an increase in central obesity. There is overwhelming evidence that high levels of antioxidant enzyme activity may be an index of protection and also reflect transcriptional upregulation of protective mechanisms in response to high levels of reactive oxygen species [35]. In vitro studies established oxidative stress as a stimuli of selenoprotein expression, including glutathione peroxidase [36]. Pendergrass and colleagues demonstrated that cardiac progenitor cells upregulated GPx gene expression after H_2_O_2_ treatment to protect the cells from injury [37]. Recently, it was found that GPx activity and GPx-3 mRNA were upregulated in patients with acute coronary syndromes compared to patients with stable coronary artery disease and healthy controls [38]. Although the regulation of GPx-3 gene is complex and poorly understood, Bierl and colleagues (2004) demonstrated that human embryonic kidney cells exposed to hypoxia for 24 h induced a three-fold increase in GPx-3 expression levels [39].

It is of note that in this study the female patients were of menopausal age. Menopause is accompanied by increased systemic oxidative stress, which is triggered by the loss of estrogen, which has been shown to have antioxidant properties in vivo and in vitro in animal models [40,41,42,43,44]. However, epidemiologic studies did not corroborate the in vivo impact of this hormone in the systemic redox balance [45]. Menopause is associated with a change in fat distribution and the lack of estrogen is linked with an inability to reduce fat mass. Changes in fat distribution in women towards a more android phenotype could have a metabolic effect of redox balance and, hence, the risk of cardio-renal disease. Recent studies have shown that the GPx-3 gene is a target gene for estrogen α receptor activation in white adipose tissue [46]. Interestingly, it was proposed that estradiol upregulate the expression of antioxidant enzymes including GPx through MAP Kinase and NFkappaB [47] that might explain the increased expression of GPx in females compared to males [48,49], but it does not explain the findings of the current study. Although, Lee et al. (2008) observed that both white and brown adipose tissues from mice have high levels of GPx-3 mRNA expression, in the presence of obesity it decreased in the white adipose tissue only [21]. While in obese male individuals, Akl and colleagues recently reported an increase in antioxidant enzymes activities and H_2_O_2_ in visceral but not subcutaneous fat [50]. It would seem that adipose tissue types respond differently to oxidative stress. These data suggest that transitional changes in body composition that occur in menopause could itself enhance oxidative stress and points to another mechanism that does not involve estrogen.

From previous studies [36,37,38,39] and ours, it is becoming apparent that the observed higher GPx-3 activity in females may be an adaptive response to increased oxidative stress loads from abdominal and visceral fat deposits. Nevertheless, a failure to sustain antioxidant defence by the production of GPx is associated with the development of cardiovascular events and renal disease progression [51,52]. Although antioxidant therapy with micronutrients has had conflicting benefit and possible harm in the general population, there has been some clear benefit in specific, high risk groups [53,54]. Our data suggest that women with type 2 diabetes at increased risk of renal and cardiovascular disease have higher GPx-3 activity, which is abrogated by increasing central fat deposits compared to men. It will be instructive to determine whether weight loss sustains GPx-3 activity and/or conversely stimulating GPx-3 activity reduces oxidative stress and cardio-renal risk in men and women with type 2 diabetes. A recent report in patients with and without diabetes with an average reduction in body weight of 25% after bariatric surgery, compared with 9% loss with lifestyle change, had a significant decrease in mRNA expression of GPx-3 in subcutaneous and visceral adipose tissue [55]. These authors did not measure GPx-3 activity but the changes in weight and gene expression support our findings and those from other cohorts that obesity and the metabolic syndrome is associated with an increase in GPx-3 activity [56].

The current study has some limitations, as it provides only a single cross-sectional measurement and cannot provide information on patients’ disease progression. In addition, we did not include patients without diabetes to corroborate the sex differences in GPx-3 activity. Furthermore, estrogen measurement was beyond the scope of this study, but such information would be beneficial for future regulatory studies of the GPx-3 enzyme. The current study sample size was adequate with good representation from Caucasian and non-Caucasian subjects who were carefully evaluated clinically to exclude clinically evident cardiovascular disease, which would have confounded the findings. This study is the first report to provide good evidence that GPx-3 activity is determined by sex and central obesity in a well-matched population in age and diabetes duration. GPx-3 activity may play a key protective role in reducing the risk of cardio-renal disease, which may be compromised in menopausal women with diabetes.

## Figures and Tables

**Figure 1 antioxidants-08-00629-f001:**
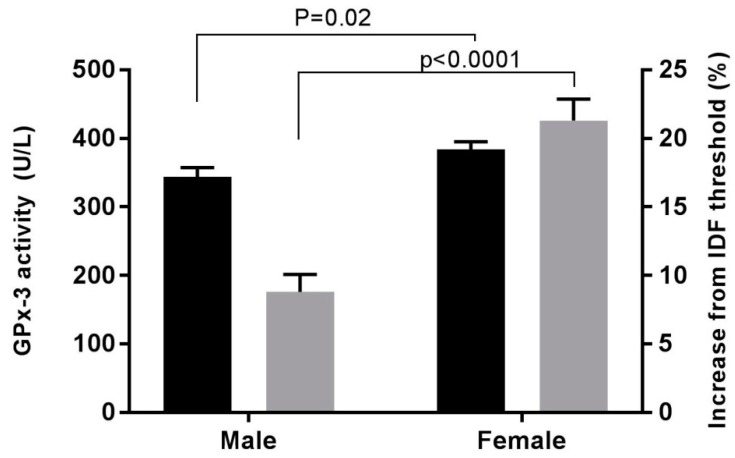
Mean (SEM) of plasma glutathione peroxidase activity (GPx-3) (black bars) and percentage increase from International Diabetes Federation (IDF) waist circumference threshold (grey bars) in male (*n* = 84) and female (*n* = 81) patients with type 2 diabetes mellitus at high risk of cardio-renal disease. Female patients, compared to their male counterparts, have an increased plasma glutathione peroxidase activity and a substantial increase from IDF waist circumference threshold percentage.

**Table 1 antioxidants-08-00629-t001:** Clinical, biochemical, and anthropometric characteristics of men and women with type 2 diabetes at high risk of developing progressive cardio-renal disease.

Demographic/Clinical Parameters	Male (*n* = 86)	Female (*n* = 85)	*p*
Age (years)	59.87 ± 8.31	61.58 ± 6.96	0.15
Diabetes duration (years)	11.06 ± 8.04	9.46 ± 6.97	0.19
Ethnicity: Caucasian (%)	49	33	
Smoking status: current-/Ex-/ or never-smoker (%)	11/46/43	6/20/74	0.0001
Systolic blood pressure (mmHg)	141.3 ± 15.36	138.77 ± 17.11	0.31
Diastolic blood pressure (mmHg)	83.08 ± 9.61	81.05 ± 8.90	0.16
HbA1c % (mmol/mol)	7.3 ± 3.8 (55.89 ± 18.26)	7.6 ± 3.8 (59.07 ± 18.32)	0.22
eGFR CKD-EPI (mL/min/1.73 m^2^)	88.76 ± 17.95	89.98 ± 16.24	0.64
Total Cholesterol (mmol/L)	3.89 ± 0.72	4.36 ± 0.92	<0.0001
Triglyceride (mmol/L)	1.62 ± 0.89	1.52 ± 1.08	0.51
HDL-cholesterol (mmol/L)	1.19 ± 0.35	1.36 ± 0.35	0.002
LDL-cholesterol (mmol/L)	1.98 ± 0.63	2.34 ± 0.78	0.001
Urinary ACR (mg:mol)	5.48 ± 14.56	1.95 ± 3.06	0.03
Bio-impedance (%)	28.99 ± 7.68	40.41 ± 8.07	<0.0001
BMI (kg/m^2^)	29.87 ± 4.88	31.02 ± 6.90	0.20
Waist Circumference (cm)	102.8 ± 11.7	101.3 ± 14.1]	0.46
Increase in waist circumference from IDF recommended threshold (%)	8.9 ± 11.7	21.3 ± 14.1	<0.0001

Data expressed as mean ± SD or median [inter-quartile range]. eGFR, estimated glomerular filtration rate; HDL, high density lipoprotein; LDL, low density lipoprotein; ACR, albumin creatinine ratio. *p* ≤ 0.05 was considered significant.

**Table 2 antioxidants-08-00629-t002:** Plasma GPx-3 activity, vitamin E, and selenium levels in men and women type 2 diabetes patients at high risk of developing progressive cardio-renal disease.

Plasma Antioxidants	Male	Female	*p*
Vitamin E:Total-Cholesterol (µmol/mmol)	8.95 ± 3.03	8.79 ± 2.54	0.72
Selenium (µmol/L)	1.25 ± 0.28	1.26 ± 0.24	0.73
GPx-3 activity (U/L)	343.3 ± 128.0	384.1 ± 99.7	0.02

Data expressed as mean ± SD, *p* ≤ 0.05 was considered significant.

**Table 3 antioxidants-08-00629-t003:** Linear regression with GPx-3 activity as the dependent variable.

Variables		β	t	*p*
**Model A**	Sex	41.0	2.28	0.02
	Waist circumference	−1.48	−2.13	0.04
**Model B**	Sex	58.0	2.53	0.01
	Bioimpedance	−1.48	−1.26	0.21

Standardized beta (β), the t test statistic (t), and the probability value (*p*).

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
