# Peer review of "Sex Differences in Glutathione Peroxidase Activity and Central Obesity in Patients with Type 2 Diabetes at High Risk of Cardio-Renal Disease"

_antioxidants, 2019, doi:10.3390/antiox8120629_

Round 1
Reviewer 1 Report
Major concerns:
1.Introduction is very short and needs to be extended. Authors need to justify why they were focused on GPx-3 activity, selenium and vitamin E as antioxidants. It is unclear why authors did not measure other antioxidant enzymes activities. Some parts from the Discussion about GPx-3 should be moved to the Introduction.
2.Discussion: authors need to state what the novelty of the study is.
Author Response
Reviewer 1.
We thank the reviewer for the comments regarding the introduction length, justification for measurement of GPx-3, and the novelty of the study. We made the following changes to our paper to address the reviewer’s comments.
In the introduction, we introduced the glutathione peroxidases family and the justification for GPx-3 analyses [Introduction, page 2, paragraph 2, Lines 53-63] In the discussion, we added the weaknesses of the study as well as the strength that includes the study’s novelty [Discussion, page 7, paragraph 4, Lines 265-273]
Reviewer 2 Report
The authors investigated the relationship of glutathione peroxidase 3 (GPx-3) activity to sex differences in central obesity in patients with type 2 diabetes. The main findings are: 1) Women had higher GPx-3 activity compared to men and an increase in waist circumference; 2) Waist circumference and BMI were negatively correlated with GPx-3 activity; 3) Waist circumference and sex were independent predictors of GPx-3 activity. Overall the study provided new information about GPx-3 activity in in patients with type 2 diabetes, however there are some concerns about the manuscript:
It would be beneficial to the manuscript if the main text was rearranged. In the current form there is very little about information about the rationale behind the study regarding GPx-3. Why the authors focused on GPx-3? Were other Gpx isoforms investigated? What about other antioxidant systems? Adding this information to introduction section and results if possible, would place the study into more context regarding the role of GPx-3 among other antioxidants in diabetes. The discussion section needs to be restructured to include more insights into the mechanisms involved in the paradox observed by the authors. It would benefit the paper if the authors expand this part of the manuscript.
Author Response
We thank the reviewer for his valuable comments and suggestions, we made the following changes to address the reviewer’s concerns.
Title: we changed the title to reflect the paper content In the introduction, we added the rational behind the focus on the analyses of GPx-3 [Introduction, page 2, paragraph 2, Lines 53-63]. In the discussion, we addressed the mechanism involved in the paradox that was highlighted in the paper [Discussion, page 7, paragraphs 1, Lines 233-234 ] and [Discussion, page 7, paragraph 2, Lines 240-246] and [Discussion, page 7, paragraph 3, Lines 258-263]

Round 2
Reviewer 2 Report
The authors provided a revised version of the manuscript including changes in the introduction and discussion section. Although some of the changes contribute to strengthen the manuscript some areas still requires modifications.
1) Introduction section still lacks information about other antioxidants enzymes and why they were not included in the study. Additionally, information about GPx-3 in the adipose tissue is only mentioned in the discussion. It would benefit the paper to add this information in the introduction.
2) Discussion section would benefit from some changes as well. The paragraph about the effects of menopause on GPx-3 levels is not clear. How low levels of estrogen would increase expression of GPx-3 in the present study? It would benefit the manuscript to include more insight into molecular mechanisms involving regulation of GPx-3 in the context of the study.
3) Conclusion should be modified to reflect the present study. The conclusion shouldn’t include data from other authors as it can lead to misinterpretations. The author conclude that “adipose GPx-3 production is determined by sex and obesity”, however GPx-3 production in adipose tissue was not addressed in the study.
4) The novelty aspects of the study should also be included in the conclusion.
5) Abstract should be changed accordingly after the alterations.
Author Response
We thank the reviewer for his/her comments, we made the following changes:
The authors provided a revised version of the manuscript including changes in the introduction and discussion section. Although some of the changes contribute to strengthen the manuscript some areas still requires modifications.
Introduction section still lacks information about other antioxidants enzymes and why they were not included in the study. Additionally, information about GPx-3 in the adipose tissue is only mentioned in the discussion. It would benefit the paper to add this information in the introduction.
In the introduction, we added information about antioxidant enzymes and the reason of the focus of the study on GPx-3 as it is mainly expressed in the kidney ( (Page 2, Paragraph 2)
Also in the introductions we mentioned briefly that GPx-3 is expressed in adipose tissue (Page2, paragraph 3, line 2). However we chose to give detailed information of GPx-3 in adipose tissue to serve the arguments in the discussion.
Discussion section would benefit from some changes as well. The paragraph about the effects of menopause on GPx-3 levels is not clear. How low levels of estrogen would increase expression of GPx-3 in the present study? It would benefit the manuscript to include more insight into molecular mechanisms involving regulation of GPx-3 in the context of the study.
Please accept our apologies for the error typo. We clarified further in the discussion section about the effect of estrogen on GPx-3 to elucidate that the sex differences that we report in this study in GPx-3 activity are not due to the estrogen effect
3) Conclusion should be modified to reflect the present study. The conclusion shouldn’t include data from other authors as it can lead to misinterpretations. The author conclude that “adipose GPx-3 production is determined by sex and obesity”, however GPx-3 production in adipose tissue was not addressed in the study. We apologies for he typo error, we rectified the mistake from adipose GPx-3 to GPx-3 activity.
4) The novelty aspects of the study should also be included in the conclusion. The novelty of the study is now clarified in the conclusion (Discussion, paragraph 3, page 8)
5) Abstract should be changed accordingly after the alterations.
Conclusion has been modified to reflect the findings of the study only.
